# Inflammatory Tumor Microenvironment in Cranial Meningiomas: Clinical Implications and Intraindividual Reproducibility

**DOI:** 10.3390/diagnostics12040853

**Published:** 2022-03-30

**Authors:** Johannes Wach, Tim Lampmann, Ági Güresir, Hartmut Vatter, Ulrich Herrlinger, Albert Becker, Marieta Toma, Michael Hölzel, Erdem Güresir

**Affiliations:** 1Department of Neurosurgery, University Hospital Bonn, 53127 Bonn, Germany; tim.lampmann@ukbonn.de (T.L.); agi.gueresir@ukbonn.de (Á.G.); hartmut.vatter@ukbonn.de (H.V.); erdem.gueresir@ukbonn.de (E.G.); 2Division of Clinical Neurooncology, Department of Neurology and Centre of Integrated Oncology, University Hospital Bonn, 53127 Bonn, Germany; ulrich.herrlinger@ukbonn.de; 3Department of Neuropathology, University Hospital Bonn, 53127 Bonn, Germany; albert_becker@uni-bonn.de; 4Institute of Pathology, University Hospital Bonn, 53127 Bonn, Germany; marieta.toma@ukbonn.de; 5Institute of Experimental Oncology, University Hospital Bonn, 53127 Bonn, Germany; michael.hoelzel@ukbonn.de

**Keywords:** intra-individual analysis, macrophages, meningioma, MIB-1, reproducibility, time to recurrence

## Abstract

The MIB-1 index was demonstrated to be significantly correlated to meningioma recurrence. However, to date, the relationship of the intraindividual course of the MIB-1 index and the growth fraction, respectively, to clinical tumor recurrence has not been demonstrated in cranial WHO grade 1 and 2 meningiomas. In the present paper, we compare the MIB-1 indices of 16 solely surgically treated primary meningiomas and their recurrent tumors regarding the course of the MIB-1 indices, time to recurrence, reproducibility and factors influencing the intraindividual MIB-1 indices. Regression analyses revealed (1) a strong intra-lab reproducibility (*r* = 0.88) of the MIB-1 index at the second versus the first operation, corresponding to a constant intrinsic growth activity of an individual meningioma, (2) a significant inverse correlation of both primary (*r* = −0.51) and secondary (*r* = −0.70) MIB-1 indices to time to recurrence, and (3) male sex, low plasma fibrinogen and diffuse CD68^+^ macrophage infiltrates contribute to an increase in the MIB-1 index. A strong intraindividual reproducibility of the MIB-1 index and a direct relationship of the MIB-1 index to the time to recurrence were observed. Individual MIB-1 indices might be used for tailored follow-up imaging intervals. Further research on the role of macrophages and inflammatory burden in the regrowth potential of meningiomas are needed.

## 1. Introduction

Meningiomas are predominantly benign neoplastic entities, which account for 36.4% of all tumors of the central nervous system (CNS) [1,2]. World Health Organization (WHO) grade 1 and 2 meningiomas make up 97–99% of all meningiomas and the surgical gross total resection is the method of choice for symptomatic meningiomas [3,4]. However, benign WHO grade 1 meningiomas can also progress after sufficient surgical treatment, and previous studies found a tumor recurrence rate of up to 47% over a 25-year long-term follow-up observation [5]. The extent of resection according to the Simpson grading system, brain invasion, mitotic activity, location, and functional status are currently used as guides to determine the appropriate adjuvant therapy, probability of recurrence and frequency of follow-up imaging [6,7,8].

Increased cellular proliferative activity is known to be an essential mechanism of oncogenesis [9]. The Molecular Immunology Borstel 1 (MIB-1) labeling index is a known immunohistochemical tool to detect nuclear components that are exclusively present during cell proliferation. The Ki-67 antigen is present in cell nuclei during the G1, S, and G2 phases of the cell division cycle as well as during mitosis. Hence, the detection of the Ki-67 antigen is a feasible technique to identify the growing fraction of neoplastic cell tissues [10,11,12]. Furthermore, numerous investigations and a recent meta-analysis have proven that the Ki-67/MIB-1 labeling index is an independent predictor of tumor progression in meningiomas [8,13,14,15]. There is currently no standardized method of MIB-1 index measurement and it has not yet been accepted as a routine classification marker for meningiomas. An international study on the reproducibility of MIB-1 index analysis showed a moderate agreement across the participating institutions. This finding might be caused by differences in the regions of tumor tissue samples, and differences in staining and scoring techniques. However, the intra-lab MIB-1 index measurement reproducibility was high [16]. The use of the MIB-1 index as a prognostic and predictive marker for progression in meningiomas has been extensively investigated in the setting of the primary diagnosis [15,17,18]. The MIB-1 index is known as a prognostic factor in both primary and recurrent breast cancers [19]. Furthermore, it has been shown that there is a strong intraindividual reproducibility of the MIB-1 index in primary glioblastomas and their recurrent tumors [20].

To date, it is unknown if the MIB-1 index can be reliably reproduced in meningiomas and if individual meningiomas exhibit an intrinsic constant proliferation activity or significant change despite cytoreductive surgical approaches. Tumor progression and regrowth potential in meningiomas are assumed to be significantly influenced by the infiltrating immune cells in the tumor microenvironment [21]. More specifically, tumor-associated macrophages comprise the majority of those immune cell types and account for 18% of all cells within meningioma tissue [22,23]. Furthermore, it is known that pro-tumoral M2 phenotype macrophages also play an essential role in terms of tumor growth potential and the immunohistochemical measurements of proliferative activity in solid tumors [21,24].

The present study aims to investigate the intraindividual reproducibility of the MIB-1 index in primary WHO grade 1 and 2 meningiomas and their local tumor recurrences, which were treated by surgical resection only. Moreover, the correlation between time to recurrence and factors (e.g., clinical, imaging, and inflammatory features) influencing the course of the MIB-1 indices in primary meningiomas and their recurrent tumors are analyzed.

## 2. Materials and Methods

### 2.1. Study Design and Patient Characteristics

Between January 2009 and July 2019, 1003 patients underwent surgery for WHO grade I and II meningiomas at the authors’ institution. A retrospective review of patient data was performed after approval by the institutional review board had been obtained. Inclusion criteria of this study were histopathologically confirmed meningiomas, intracranial localization, an age greater than 18 years, the availability of the MIB-1 index, and the presence of a recurrence at the primary resection cavity with intention to retreatment by surgery. Patients with a neurofibromatosis type 2-associated cranial meningioma or spinal meningiomas were excluded because of their differences regarding histopathology and proliferation potential [25,26]. Moreover, meningioma patients who were treated by a partial resection or biopsy (Simpson grades IV and V) were separately analyzed because those meningioma tissues do not necessarily contain the “hotspot area”, which represents the area of the neoplasm with the highest proliferative potential [27,28]. A total of 16 patients was included for the final data analysis (see Figure 1).

### 2.2. Data Recording

Clinical features including age; sex; comorbidities; Karnofsky Performance Status (KPS); tumor size; peritumoral edema; presence of multiple meningiomas; WHO grading based on an postoperative histopathological examination; immunohistochemical examinations; extent of tumor resection based on the Simpson grading system according to the European Association of Neuro-Oncology (EANO) (Simpson grades 1–3 = gross total resection, Simpson grade 4 = subtotal resection, and Simpson grade 5 = biopsy); and postoperative follow-up data were collected and entered into a computerized database (SPSS, version 27 for Mac, IBM Corp., Armonk, NY, U.S.A.) [29]. MR imaging was routinely performed within 48 h before surgery. Peritumoral edema was defined as a high signal intensity adjacent to tumors on T2-weighted MR images [30]. Tumor volume was determined using 3D semi-automatic volumetry by selecting the region of interest (Smartbrush^®^ software by Brainlab AG, Feldkirchen, Bavaria, Germany). Tumor volumes were determined by selecting the enhancing tumor mass in the T1 sequences after a gadolinium injection (see Figure 2). Volumetric analysis of the regrowth of residual meningioma tissue in Simpson grade IV or V resected tumors was performed. Meningioma regrowth was calculated by the determination of the difference between the recurrent tumor volume and residual tumor volume.

Laboratory values were recorded using the laboratory information system Lauris (version 17.06.21, Swisslab GmbH, Berlin, Germany). Venous blood was routinely collected within 24 h prior to surgery for intracranial meningiomas. These laboratory investigations were performed at constant time points, which enable a reliable analysis of progression-free survival. The routine blood examination protocol before surgery included complete blood count, kidney test, liver tests, and the coagulation profile (INR, aPTT). Plasma fibrinogen concentrations were determined using the Clauss method, which involves adding a standard and high concentration of thrombin (Dade^®^ thrombin reagent, Siemens Healthineers, Erlangen, Bavaria, Germany) to platelet-poor plasma. This fibrinogen level is determined based on a reference curve. The serum C-reactive protein values were obtained by turbidimetric immunoassays with a CRPL3 reagent (Roche, Basel, Switzerland) [31].

### 2.3. Histopathology

Histopathological grading was performed based on the 2016 WHO criteria [3]. All histopathological reports underwent repeated review to reconfirm that the diagnosis was in keeping with these requirements. Immunohistochemistry was performed in a similar workflow, as described before for paraffin-embedded biopsy tissue specimen [32,33]. The MIB-1 labeling index was determined by the usage of the following antibody: Anti-Ki67 (Clone 2B11 + PD7/26). Moreover, semiquantitative analysis and the scoring of CD68^+^ stainings using anti-CD68 antibodies to detect macrophages was performed (Clone KP1, dilution 1:1000, DAKO, Glostrup, Denmark). Meningioma specimens were investigated for the absence, focal or diffuse staining of CD68^+^ macrophages [34]. Visualization was by diaminobenzidine and the neuropathological assessment was performed by expert neuropathologists, including A.J.B. The further neuropathological workflows were as previously reported [34].

### 2.4. Follow-Up

Clinical and imaging follow-up consisted of MRI scans at 3 months after surgery, as well as on an annual basis for the following 5 years after surgery [35]. Earlier clinical and imaging appointments were scheduled in case of new or progredient worsening of neurological functions as well as radiological signs indicating tumor progression or recurrence. Recurring tumors with radio-clinical correlations, regrowing at the site of the initial surgery, were considered for analysis. The time to recurrence was defined as the period between the first surgery and the first subsequent surgical resection for recurrence at the site of the initial surgery.

### 2.5. Statistical Analysis

Data were organized and analyzed using SPSS for Mac (version 27.0; IBM Corp, Armonk, NY, USA). Kolmogorov–Smirnov (KS) tests were performed to compare the distributions. Normally distributed data are reported as the mean with the standard deviation (SD). Preoperative demographic data, imaging features, extent of resection, and histopathological features were compared regarding time to recurrence using the independent *t*-test. Age and MIB-1 index were dichotomized by the median split into <54 vs. ≥54 and <9% vs. ≥9%, respectively [36]. The intraindividual reproducibility of the MIB-1 labeling index in primary meningiomas and their recurrences was analyzed using simple linear regression analysis. Furthermore, the MIB-1 labeling index determined in the primary and recurrent meningioma tissues was also investigated regarding the correlation with time to recurrence by using Spearman’s correlation analysis and simple linear regression analyses. In the linear regression analysis, the time to recurrence was the dependent variable, while the MIB-1 labeling index determined at the primary surgery or after surgical retreatment was the independent variable. A *p*-value of <0.05 was considered statistically significant.

## 3. Results

### 3.1. Patient Characteristics

A total of 16 patients with a median age of 53.5 years (IQR 42.5–69.8) was enrolled in the present study. The study included 9 females (56.3%) and 7 males (43.7%; female/male ratio 1.29:1). The median primary tumor volume (25th–75th percentile) was 46.0 cm^3^ (22.4–72.4). The area of falx (50.0%) was the predominant location of intracranial meningiomas in the present study cohort, followed by convexity (18.8%) and spheno-orbital (12.5%) location. Multiple meningiomas were observed in 4 (25.0%) patients, and peritumoral edema was present in 11 (68.8%) patients. With regard to the extent of resection, Simpson grade I/II resections were performed in 14 patients (87.5%), whereas 2 (12.5%) patients underwent a Simpson grade III resection. Tumor classification according to the WHO classification criteria included 8 patients with grade 1 (50.0%), and 8 patients with grade 2 (50.0%). Median (IQR) MIB-1 labeling indices in primary and their corresponding recurrent meningioma were 9 (6.5–10.0) and 10 (7.3–10.0), respectively. Diffuse CD68^+^ macrophage infiltrates were observed in 31.25% of the primary meningiomas, whereas 57.1% of the recurrent meningioma tissues included diffuse CD68^+^ macrophage infiltrates. Table 1 summarizes the patient characteristics.

### 3.2. Increased MIB-1 Labeling Index

Median MIB-1 labeling indices in primary and their corresponding local recurrent meningiomas were 9.0% (6.5–10.0) and 10.0% (7.3–10.0), respectively. On neuropathological investigation, primary elevated MIB-1 labeling indices ≥ 9% were analyzed regarding the correlation with strong CD68^+^ macrophage infiltrates; 40% of primary meningiomas demonstrating an MIB-1 labeling index ≥ 9% were found to have diffuse CD68-positive macrophage infiltrates, whereas in the group of patients with primary MIB-1 labeling indices < 9%, increased CD68^+^ macrophage staining was found in only 16.7% of primary cranial meningiomas. Table 2 summarizes the results of the analyses for both primary and their local recurrent meningiomas.

### 3.3. Time to Recurrence

The mean time to recurrence in the 16 identified cases that underwent second surgery for recurrence at the primary resection cavity was 38.9 (range: 12.0–104.0) months. A comparative analysis of clinical, imaging, and histopathological features regarding the time of recurrence was performed (Table 3). Univariable analysis using independent *t*-test revealed that patients with an MIB-I labeling index ≥ 9% (determined in the recurrent tumor tissue) had a significantly shorter time to meningioma recurrence (27.6 ± 16.0 months), compared to the patients with an MIB-1 labeling index < 9% (53.6 +/− 28.8 months) (*p* = 0.04). Diffuse infiltrates of CD68^+^ macrophages were found to be associated with a shortened time-to-tumor recurrence, compared to focally infiltrating CD68^+^ macrophages (49.5 ± 27.6 vs. 15.8 ± 3.6, *p* = 0.002; see Figure 3).

Spearman’s correlation analysis was performed to determine the linear relationship between the MIB-1 labeling index at the initial diagnosis and the time to recurrence of meningiomas. MIB-1 labeling indices at the initial diagnosis are significantly and inversely correlated with the time to recurrence of cranial meningiomas (*p* = 0.045, *r* = −0.507).

Figure 4 displays a simple linear regression analysis to explain the effect of the MIB-1 labeling index at the initial diagnosis on the time to recurrence. The MIB-1 labeling index at initial diagnosis was found in a clear inverse function to the time to recurrence (standardized regression coefficient: −0.507, 95% CI: −0.801–−0.016, *p* = 0.045).

Spearman’s correlation analysis was also performed to investigate the correlation between the MIB-1 labeling index of the recurrent tumor tissue and time to recurrence of meningiomas. The MIB-1 labeling indices of the recurrent meningiomas are also significantly and inversely correlated with the time to recurrence of the cranial meningiomas (*p* = 0.003, *r* = −0.698). Figure 5 displays a simple linear regression analysis to explain the effect of the MIB-1 index determined in recurrent meningioma tissues on the time to recurrence. The MIB-1 labeling index of the recurrent meningioma was also found in a clear inverse function to the time to recurrence (standardized regression coefficient: −0.698, 95% CI: −0.886–−0.31, *p* = 0.003).

### 3.4. Intraindividual Reproducibility of the MIB-1 Labeling Index in Primary and Recurrent Cranial Meningiomas

Spearman’s correlation analysis revealed a strong positive and statistically significant correlation between the MIB-1 labeling indices of primary meningiomas and their recurrent tumor tissues (*p* < 0.001, *r* = 0.880). Figure 6 displays a simple linear regression analysis to illustrate the comparison of the corresponding MIB-1 labeling indices in primary and recurrent meningiomas. Regression analysis revealed a constant proliferation potential of the individual tumor cell population and a high intra-lab reproducibility of the intraindividual measurements (correlation coefficient: 0.880, 95% CI: 0.68–0.96, *p* < 0.001).

### 3.5. Factors Influencing the Intraindividual Course of the MIB-1 Labeling Indices in Primary and Their Corresponding Recurrent Meningiomas

A total of 9 (56.3%) patients had an identical MIB-1 labeling index at the primary and subsequent immunohistochemical examination. A total of 6 (37.5%) patients showed an increase in the MIB-1 labeling index, and a decrease in the MIB-1 labeling index was found in only 1 (6.3%) patient. The range of intraindividual differences of the MIB-1 labeling indices at the primary and subsequent analyses was (−) 2.0 – (+) 6.0. The mean difference (± SD) of the MIB-1 labeling indices between the MIB-1 labeling indices in the primary and their corresponding recurrent meningiomas was 0.81% ± 1.97. Univariable analysis using Fisher’s exact test (two-sided) and independent *t*-test was performed to investigate the potential impact of baseline clinical, imaging, laboratory, and histopathological features on the course of the MIB-1 labeling indices. Male sex, low plasma fibrinogen levels and diffuse CD68^+^ macrophage infiltrates were found to be significantly associated with an increase in the MIB-1 labeling index in recurrent tumor tissue samples. Table 4 summarizes the results of the factors that were investigated regarding their influence on the course of the MIB-1 labeling index. Figure 7 displays that diffuse CD68^+^ macrophage infiltrates contribute to increased MIB-1 labeling indices at the investigation of the primary tumors and the recurrent meningioma tissues.

### 3.6. Correlation between the Time to Recurrence, Volumetric Tumor Regrowth and Pattern of Recurrence with the MIB-1 Labeling Index in Simpson Grade IV or V Resected Meningiomas

Forty-seven patients underwent a Simpson grade IV (*n* = 45) or V (*n* = 2) resection. The median (IQR) MIB-1 labeling index in this subgroup regarding the extent of resection was 4.5% (3.0–5.0). Ten (21.3%) recurrent meningiomas were identified in this subgroup. The mean (±SD) MIB-1 labeling index in recurrent meningiomas was 8.5 (±4.7), and 4.1 (±1.9) in the patients without recurrence, respectively (*p* = 0.02). Recurrences were also analyzed regarding the pattern of regrowth. Seven (7/47; 14.9%) patients had a local recurrent meningioma, whereas only one (1/47; 2.1%) patient had a solely distant meningioma progression. Two patients (2/47; 4.3%) had a simultaneous progression of a local meningioma and a further distant meningioma. The mean (±SD) time to recurrence in Simpson grade IV or V resected meningiomas ≥ 9% was 31.2 months (±32.6), whereas, in patients with an MIB-1 labeling index < 9%, the mean time to recurrence was 68.4 months (±26.4; independent *t*-test: *p* = 0.08). Volumetric analysis of the regrowth of residual meningioma tissues in Simpson grade IV or V resected tumors was performed. The median (IQR) residual tumor volume, recurrent tumor volume, and volumetric regrowth were 11.1 cm^3^ (3.1–36.8), 35.0 cm^3^ (11.6–63.2), and 6.0 cm^3^ (4.5–27.3), respectively. The primary MIB-1 labeling index was correlated with the volumetric meningioma regrowth which was calculated by the determination of the difference between recurrent tumor volume and residual tumor volume. The mean volumetric regrowth was 30.2 cm^3^ (±30.5) in patients with an MIB-1 labeling index ≥9%, and 8.0 cm^3^ (±8.0) in patients with an MIB-1 labeling index <9%, respectively (independent *t*-test: *p* = 0.16). Re-surgery was performed on two (2/10; 20%) recurrent meningiomas of the patients who previously underwent Simpson grade IV or V resections. In both reoperated recurrent meningioma patients, the secondary MIB-1 labeling indices were higher. The further recurrences were treated by radiotherapy.

## 4. Discussion

The present investigation analyzed the intraindividual reproducibility of the MIB-1 labeling index in solely surgically treated cranial WHO grade 1 and 2 meningiomas, and their corresponding recurrences at the primary resection cavity. Furthermore, the correlation between the MIB-1 labeling indices determined at the initial and second surgery, association with time to recurrence and factors influencing the MIB-1 index were investigated. Our findings can be summarized as follows: (1) MIB-1 labeling indices of the primary meningioma and of the corresponding recurrence at the primary resection cavity strongly correlate. Hence, meningiomas exhibit a predominantly intrinsic constant proliferation activity despite cytoreductive surgical treatment. The MIB-1 assessment in meningiomas seems to be reproducible in an uniform intra-lab setting; (2) MIB-1 labeling indices of both primary and their corresponding recurrent meningioma as well as diffuse CD68^+^ macrophage infiltrates are significantly and inversely correlated with the time to recurrence of cranial WHO grade 1 and 2 meningiomas; and (3) male patients, patients having a low plasma fibrinogen level or harboring diffuse CD68^+^ macrophage infiltrates within the tumor tissue might have a progressive course of cellular proliferative potential in locally recurrent meningiomas.

Correlation and simple linear regression analysis revealed that the MIB-1 labeling indices of primary meningiomas and their local recurrent tumors are strongly correlated in a linear fashion. In the present investigation, we selectively analyzed patients who underwent only surgery without adjuvant radiation therapy and who were re-operated for a local recurrence at the primary resection cavity. A previous report analyzed Ki-67 indices in both patients treated with adjuvant radiation or not [37]. They found a significant increase in the second MIB-1 indices in patients who underwent a radiotherapy. In the group of patients without adjuvant radiation therapy, the MIB-1 indices of the primary and recurrent meningioma did not differ. However, this study has not considered essential potential confounders in the analysis of cellular proliferative potential in meningiomas, such as anatomical (e.g., skull base vs. non-skull base location) features, growth pattern (sinus invasion, brain invasion, and multiple meningiomas), and surgical extent of resection [8,26,38,39,40]. Tumor specimens of meningiomas that were surgically treated by a partial resection or biopsy (Simpson grades IV and V) do not necessarily contain the “hotspot area”, which represents the area of the tumor with the highest proliferative activity [27,28]. In our analysis of this subgroup (Simpson grades IV and V) regarding the extent of resection, the MIB-1 labeling index was not statistically significant associated with the time to recurrence. However, even in this subgroup, patients with a recurrent meningioma had a significantly higher MIB-1 labeling index compared to the patients without recurrence. Furthermore, skull-base meningiomas were found to have a significantly lower MIB-1 labeling index and the feasibility of a gross total resection might be more challenging compared to convexity or falcine meningiomas.

The MIB-1 labeling indices of the primary meningiomas as well as the MIB-1 labeling indices of their locally regrowing tumors are strongly and inversely correlated with the time to recurrence of cranial WHO grade 1 and 2 meningiomas. A significant shorter mean time-to-meningioma recurrence was observed in patients with an MIB-1 labeling index ≥ 9% in the examined primary tumor tissue specimens. This finding is in line with the results of a recent prospective trial, which identified MIB-1 labeling index as a marker for time to regrowth. This study investigated the rates of recurrence and the time to meningioma recurrence in WHO grade 1–3 meningiomas. They demonstrated that meningioma patients with an MIB-1 labeling index of 0% to 4%, 5% to 9%, and ≥10% had 2.4, 4.9, and 9.7 recurrences per 100 person years, respectively. Moreover, patients with an MIB-1 index ≥ 5% had significantly more often recurrences of meningioma within the first 2 years after surgical treatment, compared to patients with an MIB-1 index 0% to 4% [17]. Against this backdrop, it is of paramount importance to consider the MIB-1 labeling index regarding tailored surveillance imaging intervals after surgery and individual identification of patients benefiting from adjuvant treatment options (e.g., radiotherapy and radiosurgery). Adjuvant radiotherapy might be an option in those meningioma patients with an elevated MIB-1 labeling index or in those meningiomas that were subtotally or partially resected. Adjuvant radiation therapy, in addition to a prior subtotal resection, has been demonstrated to result in a reduction in the rates of recurrence [28,41,42,43,44]. Nevertheless, the majority of neuro-oncological centers still perform a watch-and-wait strategy in subtotally resected meningiomas [45].

In addition to the known association between the MIB-1 index and time to recurrence, we identified that patients harboring meningiomas with diffuse CD68^+^ macrophage infiltrates also had a significantly shorter time to recurrence, compared to patients with only focally observed CD68^+^ macrophage infiltrates. Macrophage infiltrates account for 18% of all cells in meningiomas and their amount increases as the histopathological grade of the meningioma increases [46,47]. On the immunohistochemical level, areas with elevated Ki-67/MIB-1 labeling indices were demonstrated to be overlapped by dense infiltrates of macrophages. This finding was already observed in previous institutional series of frontal skull-base meningiomas and was also demonstrated to independently influence the MIB-1 labeling index [28,35]. However, in the present intraindividual investigation of primary meningiomas and their corresponding local recurrence, we identified that an increased density of CD68^+^ in the primary investigated meningioma tissue significantly decreases the time to tumor progression compared to patients without only focally stained CD68^+^ macrophages in the primary meningioma tissue. Tumor-associated macrophages can be differentiated into a M1 or M2 phenotype. M1 phenotype macrophages have a predominantly anti-tumoral function, whereas the M2 phenotype is supposed to be a pro-tumoral macrophage [48]. Proctor et al. [21] performed an immunohistochemical analysis of 30 meningiomas. They found that more than 80% of the tumor-associated macrophages are of pro-tumoral M2 phenotype and they correlate with the meningioma size. However, we could not find an association between the tumor volumes and the semi-quantitatively determined number of macrophages in our small cohort. Furthermore, they investigated the distribution of M1 and M2 phenotype macrophages in both WHO grade 1 and 2 meningiomas. They found that WHO grade 2 meningiomas and recurrent meningiomas significantly decreased M1/M2 cell ratios compared to WHO grade 1 and primary meningiomas. Conversely, a strong inverse correlation between the density of M2 phenotype macrophages and survival rate was found in pancreas, lung, thyroid, and gallbladder cancers [24]. Hence, those findings suggest that the presence of M2 phenotype macrophages contributes to tumor growth and increased rate of recurrence. Genetic alterations are also important regarding immune responses and the polarization of macrophages. It was found that WHO grade 1 meningiomas with an AKT1 mutation predominantly had M2 macrophages, which results in a locally immunosuppressed tumor microenvironment [49].

Intraindividual change of primary and secondary MIB-1 labeling index was analyzed with consideration to age, sex, localization, growth pattern, inflammatory laboratory values, WHO grading, histopathology, and extent of resection. The male sex was associated with a significant change, and a slight increase in the MIB-1 index was observed. This simple association between the male sex and elevated MIB-1 indices was also observed in some previous investigations [50]. Kasuya et al. [51] investigated the growth rate using the MIB-1 index in a consecutive series of 342 meningioma patients. They also revealed that the male sex is an independent risk factor for an elevated MIB-1 labeling index in their logistic regression analysis. Though multiple existing studies reporting an association between the male sex and increased MIB-1 indices, the observation that recurrent cranial meningiomas in males have an accelerating proliferative potential seems to be paradoxical because of the known relationship between tumor growth and female sex hormones [52,53]. However, it must be reminded that this identified association in our study can be caused by the small sample size leading to chance findings.

The presence of CD68^+^ macrophage infiltrates in the primary tumor tissue was also significantly associated with an intraindividual increase in the MIB-1 index. The composition of immune cells infiltrating the tumor microenvironment are of paramount importance regarding tumor progression. Most of the macrophages within the infiltrates in meningiomas are polarized to the pro-tumoral M2 phenotype. However, meningiomas with chromosome 22q deletion were found to predominantly have M1 phenotype macrophages [54]. In addition to the role of the polarization of the macrophages, Han et al. [55] showed that patients with meningiomas harboring the programed death ligand-1 (PD-L1) expressing macrophages had a worse survival prognosis. Hence, targeting macrophages expressing PD-L1 might be one of the main avenues in future potential immune therapy approaches. PD-L1 expressing meningioma cells might inhibit the activation of T cells by binding to the PD-1 surface receptor of both T and B cells [56]. Due to the emerging evidence on PD-L1 expression in recurrent meningiomas, there are several ongoing trials investigating the anti-PD1 antibodies nivolumab, avelumab, or pembrolizumab in WHO grade 3 or recurrent meningiomas [57]. In a recent retrospective investigation by Tian et al. [58], 36 patients with a WHO grade 3 meningioma were analyzed regarding the prognostic value of the MIB-1 index. They demonstrated that MIB-1 labeling index > 30% is strongly associated with an increased risk of tumor progression and death in this mentioned study [58]. A literature analysis showed a range of 11% to 16.3% regarding the mean values of the MIB-1 labeling indices in WHO grade 3 meningiomas [59]. Despite several studies reporting a strong association between increased MIB-1 labeling indices and tumor recurrence or survival [58,60], there are also series that failed to demonstrate the MIB-1 labeling index as an independent marker for risk stratification in WHO grade 3 meningiomas [61,62]. WHO grade 3 meningiomas are known to be strongly infiltrated by immune cells. However, tumor cells of metaplastic or anaplastic features might show xanthomatous characteristics, which consists of CD68^+^ meningioma cells. Hence, this dilemma regarding the differentiation between macrophage infiltrates and meningioma cells may also make the determination of the MIB-1 labeling index challenging [63].

In the present study, we also found an association between the systemic inflammatory parameter plasma fibrinogen level and an increase in the MIB-1 index in the intraindividual comparison. Recently, we found that low plasma fibrinogen levels are significantly associated with an MIB-1 index ≥ 6% [64]. This inverse association between plasma fibrinogens and an increased or increasing MIB-1 index in cranial meningiomas might be explained by the potential autocrine secretion of interleukin-6 (IL-6) by human meningioma cells [65]. It was found that interleukin-6 acts as inhibitory on the growth of meningioma cells. Conversely, anti-IL-6 antibodies can enhance the tumor growth rate in meningiomas [65]. The plasma fibrinogen is linked to the interleukin-6 gene promotor, and this laboratory parameter might be induced by the IL-6 secretion of meningioma cells [66]. Hence, patients with a low plasma fibrinogen level might have a reduced autocrine IL-6 secretion by the meningioma cells, resulting in an increased tumor growth rate and proliferative potential. Furthermore, the serum C-reactive protein, which is also linked to the IL-6 gene promotor, was found to be capable of inducing the polarization of human macrophages to an M1 phenotype, and simultaneously inhibiting the transformation process to the M2 macrophages [67]. Contrary to our findings, Chen et al. [68] observed that atypical meningioma patients with an elevated preoperative fibrinogen level had a shorter progression-free survival time in a 3-year follow-up period. Consequently, the role of fibrinogen in proliferative activity in meningiomas still seems to be a paradox.

All in all, it can be summarized that the individual MIB-1 index is a reliable marker for time to recurrence, and the time intervals of follow-up imaging should be tailored using the individual MIB-1 index. The majority of the meningiomas have a constant intrinsic proliferative activity. The presence of dense CD68^+^ macrophage infiltrates also seem to be a prognostic finding regarding the time to recurrence and intraindividual increase in the MIB-1 index by inducing tumor regrowth.

The main limitation of this study is its retrospective nature and the size of the cohort. Despite a small patient cohort, the findings of this investigation reveal a novel insight into the intraindividual proliferative activity and the role of inflammatory burden. However, the differentiation of the macrophage phenotypes cannot be identified by the applied methods. Furthermore, the increased expression of CD68 can also be selectively observed in tumor cells of rare subtypes of meningiomas, such as the histiocytic or xanthomatous meningioma [62,69,70]. However, the mentioned histopathological subtypes of meningiomas were not present in this investigation. There are also pitfalls associated with the sampling of CNS tumors regarding the assessment of the MIB-1 index. In patients who underwent a subtotal resection, the sampled meningioma tissue does not necessarily contain the area with the highest proliferative potential [27]. Moreover, the MIB-1 index determination is also limited by the interobserver variability in a study analyzing a long time period. Future studies will have to reconfirm those findings by using digital image analysis systems, which allow for an objective analysis of a greater number of microscopic fields [71].

## 5. Conclusions

The present intraindividual investigation revealed that the MIB-1 index can be reliably reproduced and meningiomas seem to exhibit a constant proliferative activity despite cytoreductive surgery. Furthermore, we found a strong inverse correlation of both the primary and secondary MIB-1 labeling indices with the time to tumor recurrence in intracranial WHO grade 1 and 2 meningioma patients. Further studies need to elucidate the enhanced tumor growth in meningioma patients harboring diffuse macrophage infiltrates.

## Figures and Tables

**Figure 1 diagnostics-12-00853-f001:**
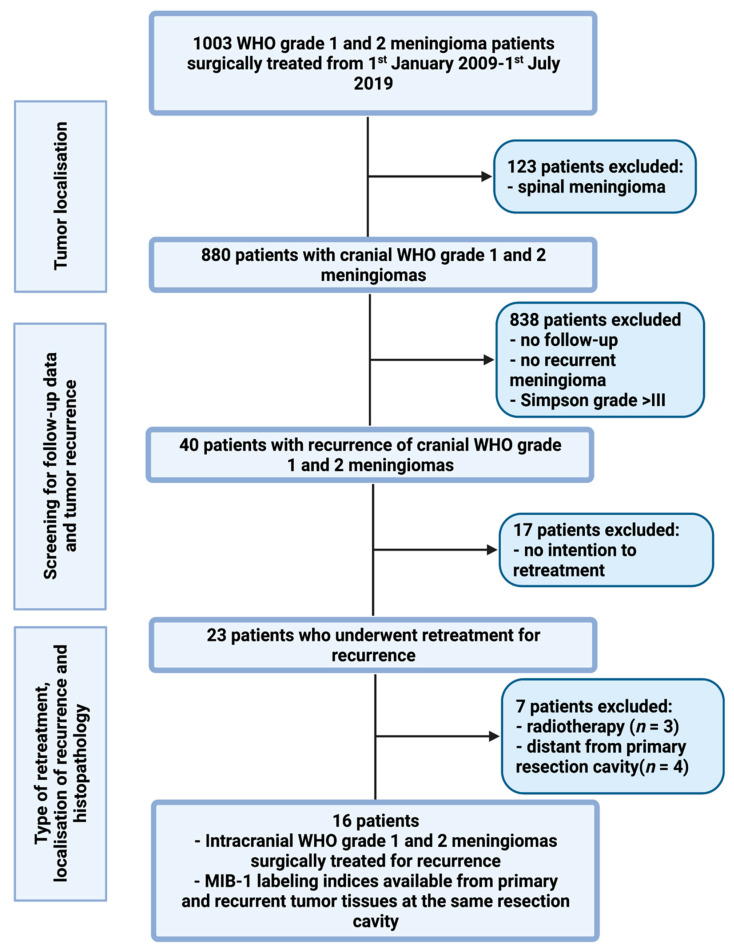
Flow chart illustrating the selection process of consecutive meningioma patients between 1 January 2009 and 1 July 2019.

**Figure 2 diagnostics-12-00853-f002:**
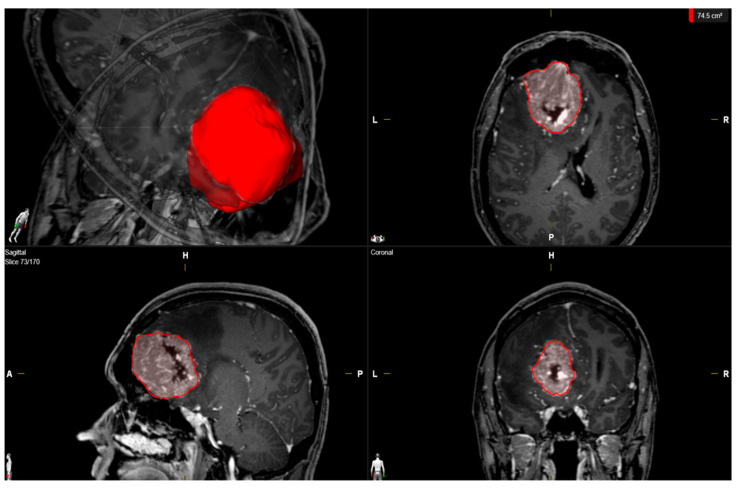
Illustrative measurements of the tumor volume (red) in a male patient with a left-sided falx meningioma using SmartBrush (Brainlab Elements, Brainlab AG, Feldkirchen, Bavaria, Germany).

**Figure 3 diagnostics-12-00853-f003:**
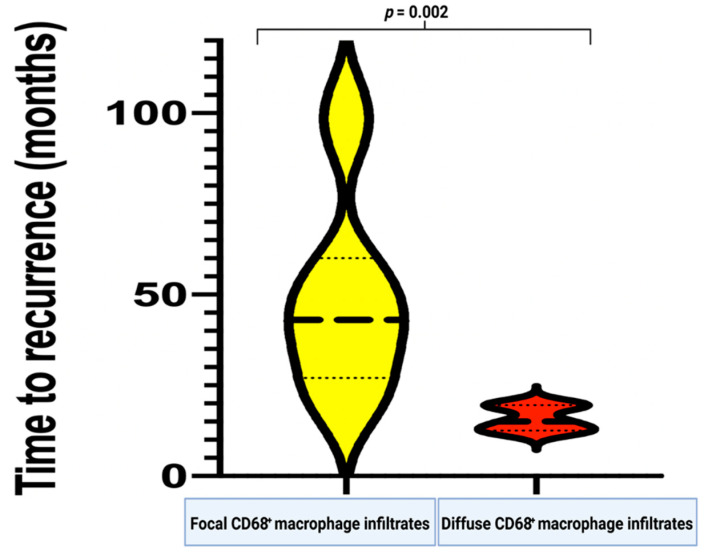
Violin plots showing the time to recurrence (in months) in relation to the extent of CD68^+^ macrophage infiltrates, which were semi-quantitatively determined (yellow constitutes the focal infiltration and red constitutes the diffuse infiltration).

**Figure 4 diagnostics-12-00853-f004:**
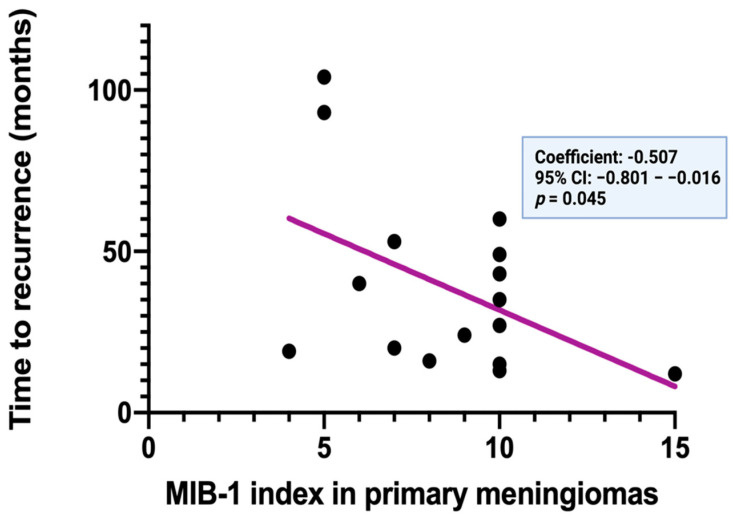
Time to recurrence (in months) in relation to the MIB−1 labeling index of 16 primary cranial meningiomas.

**Figure 5 diagnostics-12-00853-f005:**
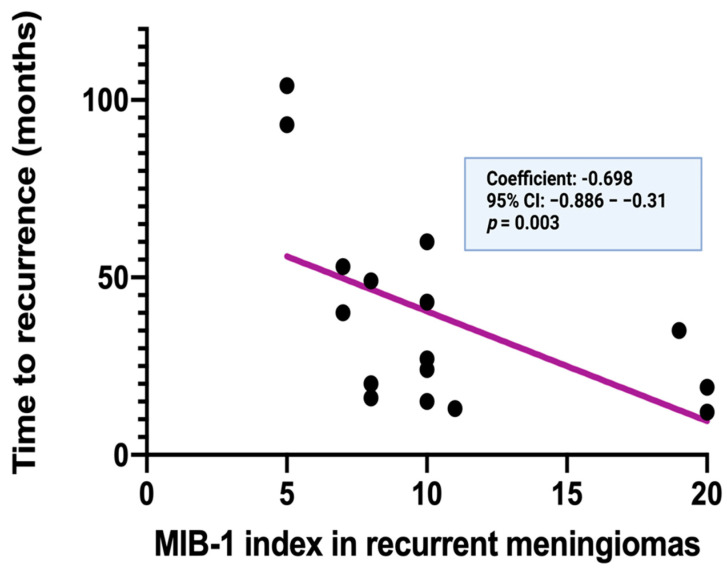
Time to recurrence (in months) in relation to the MIB−1 labeling index of the 16 corresponding recurrent cranial meningiomas.

**Figure 6 diagnostics-12-00853-f006:**
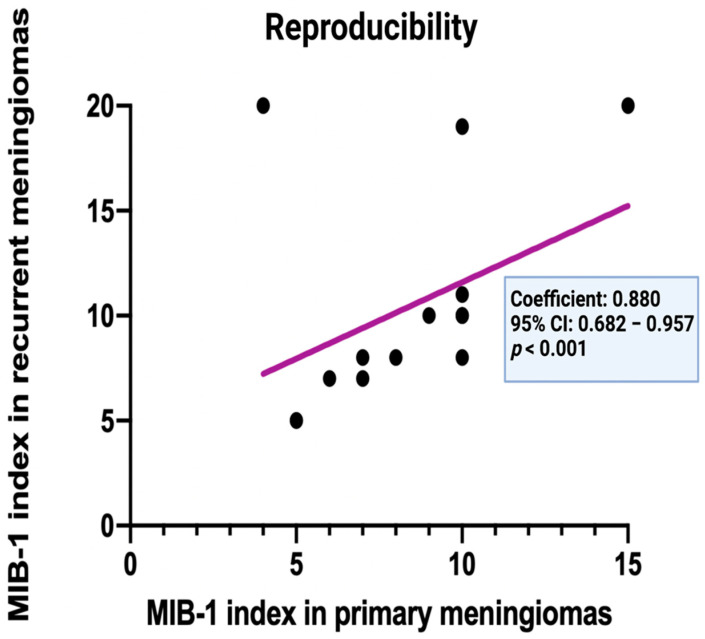
Corresponding MIB−1 index (*n* = 16) of primary and recurrent cranial meningiomas.

**Figure 7 diagnostics-12-00853-f007:**
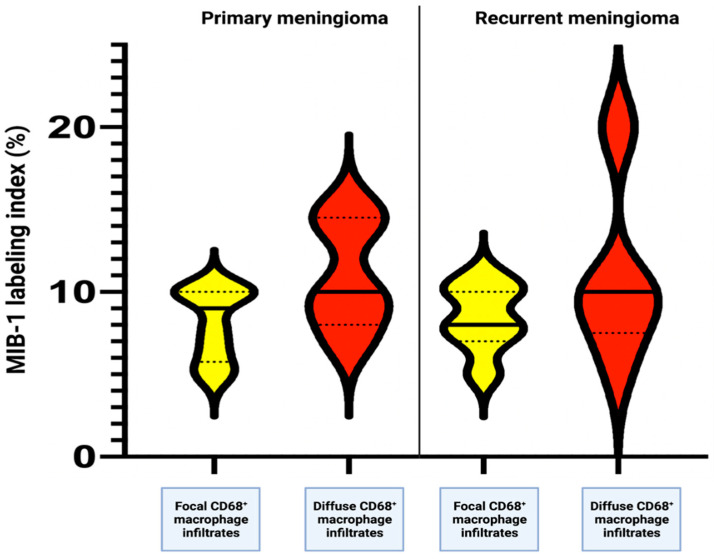
Violin plots displaying MIB-1 labeling index (*n* = 16) in patients with focal (yellow) or diffuse (red) CD68^+^ macrophage infiltrates in primary and their corresponding recurrent meningiomas.

**Table 1 diagnostics-12-00853-t001:** Patient characteristics (*n* = 16).

Median Age (IQR) (in Year)	53.5 (42.5–69.8)
Sex	
Female	9 (56.3%)
Male	7 (43.7%)
Median preoperative KPS (IQR)	85 (80–90)
Tumor location	
Falx	8 (50.0%)
Convexity	3 (18.8%)
Spheno-orbital	2 (12.5%)
Sphenoid wing	1 (6.25%)
Olfactory groove	1 (6.25%)
Intraventricular	1 (6.25%)
Multiple meningiomas	4 (25.0%)
Median baseline tumor volume (cm^3^, 25th–75th percentile)	46.0 (22.4–72.4)
Median tumor volume of recurrent meningiomas (cm^3^, 25th–75th percentile)	6.13 (4.0–21.5)
Peritumoral edema	11 (68.8%)
Sinus invasion	3 (18.8%)
Brain invasion	2 (12.5%)
Simpson grade	
Simpson grades I and II	14 (87.5%)
Simpson grade ≥ III	2 (12.5%)
WHO grade	
WHO grade 1	8 (50.0%)
WHO grade 2	8 (50.0%)
Median primary MIB-1 index (IQR)	9 (6.5, 10.0)
Median MIB-1 index in recurrent tumor (IQR)	10 (7.3, 10.0)
Median primary mitotic count (IQR)	4 (2.3. 6.0)
Median mitotic count in recurrent tissue (IQR)	3 (2.0, 6.0)
CD68^+^ macrophage infiltrates (Primary tumor)	
Focal	11 (68.75)
Diffuse	5 (31.25%)
CD68^+^ macrophage infiltrates (Recurrent tumor) [available in 14 patients]	
Focal	6 (42.9%)
Diffuse	8 (57.1%)

**Table 2 diagnostics-12-00853-t002:** MIB-1 labeling index and CD68-positive macrophage infiltrates.

*Primary Immunohistochemical Investigation*	CD68^−^ Cells	CD68^+^ Cells	*p*-Value
MIB-1 labeling index < 9%	5 (83.3%)	1 (16.7%)	0.59
MIB-1 labeling index ≥ 9%	6 (60.0%)	4 (40.0%)	
*Immunohistochemical investigation in recurrent meningiomas* [available in 14 patients]			
MIB-1 labeling index < 10%	3 (50.0%)	3 (50.0%)	0.99
MIB-1 labeling index ≥ 10%	3 (37.5%)	5 (62.5%)	
“−” constituting no to focal positive staining; “+” constituting elevated positive staining			

**Table 3 diagnostics-12-00853-t003:** Association between clinical, imaging, histopathological features, and the time to recurrence. Significant results (*p* < 0.05) are shown in bold type.

Variable	Mean (Months)	+/− SD (Months)	*p*-Value
**Age**			0.98
≥54	39.3	17.1	
<54	38.8	31.2	
**Sex**			0.07
Male	24.9	12.4	
Female	49.9	31.9	
**Tumor volume**			0.49
<46 cm^3^	33.9	16.5	
≥46 cm^3^	44.0	36.3	
**Peritumoral edema**			0.36
Present	32.7	20.2	
Not present	43.2	34.8	
**Multiple meningiomas**			0.81
Present	38.0	18.6	
Not present	35.6	25.4	
**Tumor localization**			0.16
Skull base	57.5	44.1	
Non-skull base	32.8	15.9	
**Sinus invasion**			0.34
Present	28.4	12.8	
Not present	36.5	27.3	
**Brain invasion**			0.74
Present	41.8	25.6	
Not present	38.0	28.3	
**Simpson grade**			0.37
Simpson grades I and II	36.5	25.0	
Simpson grade ≥ III	56.0	52.3	
**WHO grade**			0.48
1	44.0	35.8	
2	33.9	17.6	
**Mitotic count (primary meningioma)**			0.82
≥4	40.6	27.3	
<4	37.3	29.9	
**MIB-I index (primary meningioma)**			**0.04**
≥9%	27.6	16.0	
<9%	53.6	28.8	
**CD68^+^ macrophage infiltrates (Primary meningioma)**			**0.002**
Focal	49.5	27.6	
Diffuse	15.8	3.6	

**Table 4 diagnostics-12-00853-t004:** Association between the baseline clinical, imaging, histopathological features and intra-individual MIB-1 labeling index differences in primary and recurrent meningiomas. Significant results (*p* < 0.05) are shown in bold type.

Variable	Smaller or Equal MIB-1 Index (*n* = 10)	Increased MIB-1 Index (*n* = 6)	*p*-Value
**Age**			0.63
≥54	5	2	
<54	5	4	
**Sex**			**0.035**
Male	2	5	
Female	8	1	
**Tumor volume**			0.99
<46 cm^3^	5	3	
≥46 cm^3^	5	3	
**Peritumoral edema**			0.99
Present	8	3	
Not present	3	2	
**Multiple meningioma**			0.23
Present	4	0	
Not present	6	6	
**Tumor localization**			0.60
Skull base	2	2	
Non-skull base	8	4	
**Sinus invasion**			0.06
Present	0	3	
Not present	9	4	
**Brain invasion**			0.99
Present	0	2	
Not present	5	9	
**Primary MIB-1 index**			0.33
≥9%	8	3	
<9%	3	4	
**Primary mitotic count**			0.99
≥4	5	3	
<4	5	3	
**CD68^+^ macrophage infiltrates**			**0.001**
Focal	10	1	
Diffuse	0	5	
**Simpson grade**			0.63
Simpson grades I and II	9	5	
Simpson grade ≥ III	1	1	
**WHO grade**			0.61
1	6	2	
2	4	4	
**Mean serum C-reactive protein level (+/− SD)**	2.73 +/− 3.35	1.02 +/− 1.19	0.30
**Mean plasma fibrinogen level (+/− SD)**	3.67 +/− 0.73	2.23 +/− 0.38	**0.02**

## Data Availability

The data presented in this study are available on request from the corresponding author. The data are not publicly available due to privacy and ethical restrictions.

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
