# Peer review of "Inflammatory Tumor Microenvironment in Cranial Meningiomas: Clinical Implications and Intraindividual Reproducibility"

_diagnostics, 2022, doi:10.3390/diagnostics12040853_

Round 1
Reviewer 1 Report
Dear Authors, I have read with interest and pleasure your paper which has fascinated me very much. This is a retrospective cohort study in which 16 patients with meningioma (grade I / II according to WHO 2016) who underwent local regression are analyzed. Patients with other variables are excluded (grade III mengioma, radiotherapy, adjuvant medical therapy, etc.). The Authors have highlighted a well known and important problem regarding the repoducibility of the MIB-1 index (Ki-67 +) in the prediction of potential recurrence of meningiomas, correlating it, by simple linear regression, to the diffuse or focal infiltration of macrophages. tissue (CD68 +). Their results found a good agreement between the MIB-1 labeling index and the risk of local recurrence, as well as an inverse correlation with macrophages: it would seem that increasing macrophage infiltration (CD68 +) is associated with a reduced time to tumor recurrence, as well as an increase in the Ki67 + index correlates directly with the risk of relapse. In the discussion section, a reasoning is properly conducted in the light of previous works that had analyzed these indices in cohorts of patients with other variables (for example radiotherapy or other). I think it is a well written and clear paper. I do not see any particular problems, except that perhaps it would have been more correct to conduct the analysis following the latest WHO 2021 classification, but I do not think that is a reason to reject the work that remains valid. I would suggest to the authors to integrate in the discussion section a small paragraph that talks about grade III meningiomas, with some data on how KI67 + and CD68 + behave in such cases.In table 1 there is an error: "canial": please correct in "cranial"
Author Response
Dear Reviewer
Thank you for thoroughly reviewing our revised manuscript and the comments, which will allow us to improve it to a better scientific level and make it more understandable to the readership.
In the following we would like to respond to the remarks:
The reviewer is absolutely right that WHO grade 3 meningiomas have a special role if the MIB-1 labeling index or the macrophage infiltrates are investigated. Despite the present investigation was focused on WHO grade 1 and 2 meningiomas, this important issue was integrated in the section discussion. Due to the low incidence of WHO grade 3 meningiomas, the present evidence regarding the prognostic role of MIB-1 index in the risk stratification of recurrence is predominantly based on investigations analyzing WHO grade 1 and 2 meningiomas. For instance, a meta-analysis [1] investigating the prognostic value of MIB-1 index in meningiomas included only two studies which exclusively analyzed MIB-1 labeling indices in WHO grade 3 meningiomas. However, a more recent retrospective study by Tian et al. [2] analyzed 36 WHO grade 3 meningiomas with regard to the prognostic value of MIB-1 labeling index. This investigation revealed a MIB-1 labeling index >30% as a strong predictor of meningioma progression and survival. Mean MIB-1 labeling indices in WHO grade 3 meningiomas were reported to range between 11 and 16.3% [3]. In contrast to the studies supporting the prognostic importance of the MIB-1 labeling index in WHO grade 3 meningiomas [2, 4], there are also investigations which could not identify MIB-1 labeling index as a variable to be associated with recurrence in WHO grade 3 meningiomas [5, 6]. Furthermore, the differentiation between meningioma cells and macrophage infiltrates might be more challenging in these cases due to potential xanthomatous differentiation by CD68+ meningioma cells [7]. However, future large multicentric cohort trials are needed to sufficiently validate the prognostic utility of MIB-1 labeling index in WHO grade 3 meningiomas.
References
- Liu, N.; Song, S.Y.; Jiang, J.B.; Wang, T.J.; Yan, C.X. The prognostic role of Ki-67/MIB-1 in meningioma: A systematic review with meta-analysis. Medicine (Baltimore). 2020, 99(9), e18644
- Tian, W.; Liu, J.; Zhao, K.; Wang, J.; Jian, W.; Shu, K.; Lei, T. Analysis of Prognostic Factors of World Health Organization Grade III Meningiomas. Fron Oncol. 2020, 10, 593073
- Perry, A.; Stafford, S.L.; Scheithauer, B.W.; Suman, V.J.; Lohse, C.M. Meningioma grading: an analysis of histologic parameters. Am J Surg Pathol. 1997, 21(12), 1455-65
- Bruna, J.; Brell, M.; Ferrer, I.; Gimenez-Bonafe, P.; Tortosa, A. Ki-67 proliferative index predicts clinical outcome in patients with atypical or anaplastic meningioma. Neuropathology. 2007, 27(2), 114-120
- Shan, B.; Zhang, J.; Song, Y.; Xu, J. Prognostic factors for patients with World Health Organization grade III meningioas treated at a single center. Medicine (Baltimore). 2017, 96(26), e7385
- Zhu, X.; Xie, Q.; Zhou, Y.; Chen, H.; Mao, Y.; Zhong, P.; Zheng, K.; Wang, Y.; Wang, Y.; Xie, L.; Zheng, M.; Tang, H.; Wang, D.; Chen, X.; Zhou, L.; Gong, Y. Analysis of prognostic factors and treatment of anaplastic meningioma in China. J Clin Neurosci. 2015, 22(4), 690-695
- Ishida, M.; Fukami, T.; Nitta, N.; Iwai, M.; Yoshida, K.; Kagotani, A.; Nozaki, K.; Okabe, H. Xanthomatous meningioma: a case report with review of the literature. Int J Clin Exp Pathol. 2013, 6(10), 2242-6
Reviewer 2 Report
Through this article, J. Wach et al evaluated the intraindividual MIB-1 index correlation to meningioma tumor recurrence for 16 patients solely treated with surgical for their primary & recurrent tumors. Their observation shows a good correlation between MIB-1 index of primary and secondary tumor, inverse correlation of these indexes to time to recurrence and contribution of sex, plasma fibrinogen and CD68+ M2 macrophage distribution to MIB-1 index.
Overall, their study supports strongly the usefulness of MIB-1 index for prediction of meningioma patient prognosis, and as they suggested, follow-up monitoring and adjuvant treatment regimen.
Given the significant impact of their study, I suggest the acceptance of their article, following several clarifications below:
- When the authors mention the differentiation of >=9% and <9% MIB-1 index, is the 9% based of cell counting for cells positive with anti-Ki67 staining? Were identification of cell type performed for this positive cell subpopulation?
- Related to above, please include the representative IHC result for such MIB-1 index differentiation.
- The authors excluded tissue samples from meningioma patients which underwent partial resection as they may not contain the hotspot area. I suggest the authors to consider evaluating such sample, whether there is any correlation between MIB-1 index with distance of location from original intracranial surgery site
Author Response
Dear Reviewer
Thank you for thoroughly reviewing our revised manuscript and the comments, which will allow us to improve it to a better scientific level and make it more understandable to the readership.
In the following we would like to respond to the remarks:
The reviewer is absolutely right that cell counting in meningiomas might be challenging and may be significantly influenced by the large amount of tumor-associated macrophages in meningiomas which comprise the majority of immune cell types and approximately account for 18% of all cells within meningiomas [1, 2]. Hence, the cell populations and areas of increased or normal MIB-1 labeling indices were further investigated using CD68 staining. We created a novel section “3.2 Increased MIB-1 labeling index” in the chapter results to provide a clearer understanding of the immunohistochemical results. Furthermore, we created a novel table 2 summarizing the neuropathological investigation. On histopathological examination, primary increased MIB-1 labeling indices ≥9% were analyzed regarding the correlation with strong CD68+ macrophage infiltrates: 40% of primary meningiomas showing an MIB-1 labeling index ≥9% were identified to be diffusely infiltrated by CD68-positive macrophages, whereas in the group of patients with an MIB-1 labeling index <9% increased CD68+ macrophage infiltrates were found in only 16.7% of primary cranial meningiomas.
We have further implemented the analysis of patients who underwent either a Simpson grade IV or V resection in a newly created section “3.6 Correlation between time to recurrence and pattern of recurrence with MIB-1 labeling index in Simpson grade IV or V resected meningiomas”. Forty-seven patients underwent a Simpson grade IV (n = 45) or V (n = 2) resection. Median (IQR) MIB-1 labeling index in this subgroup regarding extent of resection was 4.5% (3.0-5.0). 21.3% (10/47) of partially or biopsied meningioma patients suffered from a tumor recurrence. However, only two patients in this subgroup underwent a second surgery for recurrent meningioma. The residual 8 recurrent meningioma patients underwent a radiotherapy for recurrence. Hence, the intraindividual analysis in this subgroup is exclusively based on the findings of the neuropathological investigation of the primary meningioma tissue. We analyzed the intraindividual time to recurrence, pattern of recurrence and volumetric tumor regrowth (in cm3) in correlation with the primary MIB-1 labeling index. Mean (+/- SD) time to recurrence in Simpson grade IV or V resected meningiomas ≥9 % was 31.2 months (+/- 32.6), and in patients with a MIB-1 labeling index <9% mean time to recurrence was 68.4 months (+/- 26.4), respectively (independent t-test: p = 0.08). Seven (7/47; 14.9%) patients had a local recurrent meningioma, whereas only one (1/47; 2.1%) patient had a solely distant meningioma progression. Two patients (2/47; 4.3%) had a simultaneous progression of a local meningioma and further distant meningioma. Primary MIB-1 labeling index was correlated with the volumetric meningioma regrowth which was calculated by the difference between recurrent tumor volume and residual tumor volume. Mean volumetric regrowth was 30.2 (+/- 30.5) in patients with a MIB-1 labeling index ≥9%, and 8.0 (+/- 8.0) in patients with a MIB-1 labeling index <9%, respectively (independent t-test: p = 0.16). Re-surgery was performed in two (2/10; 20%) recurrent meningiomas of the patients who primary underwent Simpson grade IV or V resection. In both reoperated recurrent meningioma patients the secondary MIB-1 labeling indices were higher. The further recurrences were treated by radiotherapy. Furthermore, those additional results were also included in the section discussion. MIB-1 labeling indices were not significantly associated with time to recurrence in this subgroup of either Simpson grade IV or V resected meningiomas which might be explained by both the small sample size and the sampling bias regarding the “hotspot area reflecting the maximum proliferative activity” resulting in an underestimation of the proliferative potential [3]. Furthermore, immunohistochemical areas with increased Ki-67/MIB-1 labeling indices were found to be overlapped by dense infiltrates of macrophage. This observation was already identified in previous institutional investigations of frontal skull base meningiomas and was also demonstrated to independently influence the MIB-1 labeling index in all sporadic meningiomas (convexity, skull base, spine) [4, 35]. Nevertheless, in the present study analyzing primary meningiomas and their corresponding local meningioma recurrence we newly identified that an increased density of CD68-positive cells in the primary examined meningioma tissue significantly decreases the time to meningioma progression compared to patients without or only focally stained CD68-positive macrophages in the primary meningioma tissue. Hence, this finding might inform the design of several future and ongoing trials investigating immunotherapies with antibodies (e.g., nivolumab, avelumab, pembrolizumab) modulating inflammatory immunoreactive milieu in meningiomas [6].
References
- Domingues, P.H.; Teodósio, C.; Ortiz, J.; Sousa, P.; Otero, A.; Maillo, A.; Bárcena, P.; García-Macias, M.C.; Lopes, M.C.; de Oliveira, C.; Orfao, A.; Tabernero, M.D. Immunophenotypic identification and characterization of tumor cells and infiltrating cell populations in meningiomas. Am J Pathol. 2012, 181(5), 1749-61.
- Asai, J.; Suzuki, R.; Fujimoto, T.; Suzuki, T.; Nakagawa, N.; Nagashima, G.; Miyo, T.; Hokaku, H.; Takei, A. Fluorescence automatic cell sorter and immunohistochemical investigation of CD68-positive cells in meningioma. Clin Neurol Neurosurg. 199, 101(4), 229-34.
- Coons, S.W.; Johnson, P.C. Regional heterogeneity in the proliferative activity of human gliomas as measured by the Ki-67 labeling index. J Neuropathol Exp Neurol. 1993, 52(6), 609-18
- Wach, J.; Lampmann, T.; Güresir, Á.; Vatter, H.; Herrlinger, U.; Becker, A.; Cases-Cunillera, S.; Hölzel, M.; Toma, M.; Güresir, E. Proliferative Potential, and Inflammatory Tumor Microenvironment in Meningioma Correlate with Neurological Function at Presentation and Anatomical Location—From Convexity to Skull Base and Spine. Cancers2022, 14, 1033
- Schneider, M.; Borger, V.; Güresir, A.; Becker, A.; Vatter, H.; Schuss, P.; Güresir, E. High Mib-1-score correlates with new cranial nerve deficits after surgery for frontal skull base meningioma. Neurosurg Rev. 2021, 44(1), 381-387
- Garzon-Muvdi, T.; Bailey, D.D.; Pernik, M.N.; Pan, E. Basis for Immunotherapy for Treatment of Meningiomas. Front Neurol. 2020, 11, 945.